# Update on *Stachybotrys chartarum*—Black Mold Perceived as Toxigenic and Potentially Pathogenic to Humans

**DOI:** 10.3390/biology11030352

**Published:** 2022-02-23

**Authors:** Mariusz Dyląg, Klaudyna Spychała, Jessica Zielinski, Dominik Łagowski, Sebastian Gnat

**Affiliations:** 1Department of Mycology and Genetics, Faculty of Biological Sciences, University of Wroclaw, 51-148 Wroclaw, Poland; 2Student Scientific Circle (SKN Mykobiota), Faculty of Biological Sciences, University of Wroclaw, 51-148 Wroclaw, Poland; 299130@uwr.edu.pl; 3Department of Oral Health Sciences, Medical University of South Carolina, Charleston, SC 29425, USA; zielinsk@musc.edu; 4Department of Veterinary Microbiology, Faculty of Veterinary Medicine, University of Life Sciences in Lublin, 20-950 Lublin, Poland; dominik.lagowski@up.lublin.pl (D.Ł.); sebastian.gnat@up.lublin.pl (S.G.)

**Keywords:** black toxic mold, biology, ecology, pathogenicity, mycotoxins, clinical implications

## Abstract

**Simple Summary:**

Among the 123 currently accepted species of the genus *Stachybotrys*, *S. chartarum* consistently deserves special attention. This one is the most frequently isolated species of the aforementioned genus and one of the world’s 10 most hazardous fungi. *S. chartarum* owes its notoriety to the secondary metabolites that are released in the host environment. Interaction of these metabolites with mucous membranes of the respiratory or digestive tract and with the skin can induce necrotic changes and even life-threatening pulmonary or gastrointestinal hemorrhage. Pulmonary hemorrhage was of major concern to the public when, in the period of 1993–1998, this ailment was identified in 138 infants in Cleveland, Ohio, USA, among which 12 cases were fatal. Since this first documented outbreak, the number of publications dealing with the toxigenic properties of this fungus, and even its potential pathogenicity, has rapidly grown. In this comprehensive review we present the most updated information about taxonomy, macro- and micromorphology, biology and ecology of this fungus. Within this work, we focus on the factors that prompted this fungus to be extremely dangerous for people and animals living in moldy conditions, as well as antifungals efficient in its eradication from indoor environments.

**Abstract:**

In nature, there are many species of fungi known to produce various mycotoxins, allergens and volatile organic compounds (VOCs), as well as the commonly known etiological agents of various types of mycoses. So far, none of them have provoked so much emotion among homeowners, builders, conservators, mycologists and clinicians as *Stachybotrys chartarum*. This species compared to fungi of the genera *Fusarium* and *Aspergillus* is not as frequently described to be a micromycete that is toxigenic and hazardous to human and animal health, but interest in it has been growing consistently for three decades. Depending on the authors of any given review article, attention is focused either on the clinical aspects alongside the role of this fungus in deterioration of biomaterials, or aspects related to its biology, ecology and taxonomic position. On the one hand, it is well established that inhalation of conidia, containing the highest concentrations of toxic metabolites, may cause serious damage to the mammalian lung, particularly with repeated exposure. On the other hand, we can find articles in which authors demonstrate that *S. chartarum* conidia can germinate and form hyphae in lungs but are not able to establish an effective infection. Finally, we can find case reports that suggest that *S. chartarum* infection is linked with acute pulmonary hemorrhage, based on fungal structures recovered from patient lung tissue. New scientific reports have verified the current state of knowledge and note that clinical significance of this fungus is exceedingly controversial. For these reasons, understanding *S. chartarum* requires reviewing the well-known toxigenic features and harmful factors associated with this fungus, by gathering the newest ones into a coherent whole. The research problem related to this fungus seems to be not overly publicized, and there is still a demand to truthfully define the real threats of *S. chartarum* and phylogenetically related species. The most important problem, which should be fully elucidated as soon as possible, remains the clarification of the pathogenicity of *S.* *chartarum* and related species. Maybe it is urgent time to ask a critical question, namely what exactly do we know 28 years after the outbreak of pulmonary hemorrhage in infants in Cleveland, Ohio, USA most likely caused by *S. chartarum*?

## 1. Introduction

People spend approximately 90% of their life indoors inhaling about 15 m^3^ of ambient air daily [1,2]. Paradoxically, the exposure to indoor air pollution might be stronger than exposure to the impurities of the outdoor air [1,3,4]. Fungal propagules can enter the living space of a building from the outside air through opened windows or doors [5]. It is commonly known that many respiratory symptoms are associated with staying in homes that have humidity problems and the presence of molds on various building materials [3,6]. Poor indoor air quality can cause different occupational diseases, many of which can be associated with toxigenic fungi [7,8]. Since 1993, when *Stachybotrys chartarum* was discovered as an etiological agent directly related with acute pulmonary hemorrhage in infants from Cleveland, Ohio, USA [9,10], the interest in this black fungus systematically grew. *S. chartarum* is not only a damp building-related fungus found under special conditions, this micromycete fungus is much more common than it might be assumed, and is favored by relatively high humidity and cellulose-rich materials [11]. Taking this into account, it seems justified to explore the topics related to this fungus in order to consciously and properly protect houses and public utility rooms against its occurrence. After many changes, the current taxonomic position of this black fungus was established [12,13], which was certainly due to new data based on the whole-genome sequencing of *S. chartarum* and *S. chlorohalonata* strains [14]. Similarly, some inconsistencies in the descriptions of the micromorphology of this fungus were resolved, such as the fact that all the strains of *Stachybotrys* genus produce single-celled conidia [15]. Moreover, actualized data on the range of pH, temperature and water activity factor (a_w_) allow us to precisely define the conditions under which growth of this fungus may occur [16,17,18], which in turn are crucial to prevent biodeterioration processes. Actually, it is commonly known that *S. chartarum* as a tertiary colonizer requires a_w_ > 0.9 and relative humidity (RH) > 90% to grow [19,20]. People can be exposed to this fungus via dermal contact, ingestion and inhalation. *S. chartarum* is capable of producing mycotoxins, which can be divided into three structural groups, macrocyclic trichothecenes (MCTs, e.g., satratoxins), atranones and immunosuppressive phenylspirodrimanes (PSDs) [21,22]. Among the most recently isolated, three new dolabellane-type diterpenoids and three new atranones showed interesting antimicrobial properties [23]. It is commonly known that most of the mycotoxins related with this species are very harmful for humans. It should be noted, however, that the Centers for Disease Control and Prevention (CDC), due to the lack of complete documentation of pulmonary hemorrhage cases described in the years 1993–1996, does not allow for the conclusion that *S. chartarum* unequivocally was the etiological agent for this ailment in infants [24]. However, it is impossible not to notice that 91% of the 52 cases of infant pulmonary hemorrhage described so far have been associated with the presence of *S. chartarum* in the homes of the studied patients [13,25]. Due to the fact that *S. chartarum* is one of the world’s 10 most feared fungi [26], and the problematic issues raised in this paper have not yet been explained, it seems justified to pay attention to this black mold.

## 2. Taxonomic Position

*Stachybotrys chartarum* (Ehrenb.) S. Hughes (1958) [12,13,27] was isolated for the first time by Corda in 1837 from wallpaper in a house in Prague (Czech Republic) and then described as *Stachybotrys atra* [12,13,28]. Current taxonomic position and known synonyms of this fungus are given in Table 1.

This species belongs to the genus *Stachybotrys*, family Stachybotryaceae, order Hypocreales, subclass Hypocreomycetidae, class Sordariomycetes, subphylum Pezizomycotina, phylum Ascomycota, subkingdom Dikarya and kingdom Fungi [30]. *S. chartarum* is also known by 132 synonyms (Table 1). Currently, the name *S. chartarum* is commonly accepted and used by mycologists worldwide [12,13,15]. According to Corda’s description from 1837, the conidia of this species were to be two chambered. This finding was controversial and finally resolved by Bisby [12,13,15,31]. It is now commonly known that all strains of *Stachybotrys* genus produce single-celled conidia [15,32]. The species *Stachybotrys junnanensis* described by Kong [33], *Stachybotrys chlorohalonata* described by Andersen et al. [34,35] and also *Stachybotrys eucylindrospora* described by Wiederhold et al. [36] are very similar to *S. chartarum* and quite difficult to distinguish. Moreover, the variation in the size and shape of phialides and conidia makes it difficult to identify *S. chartarum*. Due to the fact that color and ornamentation of conidia change with age [15,37], it is easy to misidentify *S. chartarum* for such species as *S. chlorohalonata*, *S. yunnanensis*, *Stachybotrys albipes* and *Stachybotrys elegans*. For this reason, molecular biology techniques are an alternative solution enabling unambiguous identification of species of *Stachybotrys* taxa [38,39,40,41,42,43], especially those recently established based on fast and reliable MALDI-TOF MS identification method [44,45]. Sometimes, however, even well-designed specific primers used in real-time PCR procedure do not differentiate *S. chartarum* from *S. chlorohalonata* or *S. yunnanensis* [15]. It is believed that the concept of the species *S. chartarum* is still broad [15]. Since the whole genomes of three *S. chartarum*, one *S. chlorohalonata* and a few other *Stachybotrys* spp. strains have been sequenced, it has been possible to show differences in the context of secondary metabolites produced at the genotype level [46]. Comprehensive analyses of genomes of three *S. chartarum* and one *S. chlorohalonata* strains revealed two mutually exclusive toxin chemotypes, one producing satratoxins (strains IBT 40293 and IBT 7711), and the second one synthesizing atranones (strains IBT 40285 and IBT 40288). The recent studies clearly explain why a strain combining the simultaneous ability to synthesize satratoxins and atranones has never been observed [14]. Apart from the differentiation at the level of strain metabolomes, the genus *Stachybotrys* currently includes 12 species and is characterized by conidiophores branching at the basal septum and the presence of thick-walled phialoconidia that are usually sticky and ornamentated [12]. Currently, it is also well known that *Memnoniella* is a synonym of *Stachybotrys* [13]. Phylogenetic studies by Castlebury et al. [47] and Summerbell et al. [48] allow for the distinction of three genera, *Myrothecium*, *Peethambara* and *Stachybotrys*, which constitute a monophyletic lineage distinct from the other families within the *Hypocreales*. Later, extensive studies based on multi-locus molecular phylogenetic analyses revealed that the genera *Myrothecium* and *Stachybotrys* are polyphyletic. Furthermore, this led to the introduction of 13 new genera with myrothecium-like morphology and 8 new genera with stachybotrys-like morphology [12]. The family of Stachybotryaceae also includes other anamorphic genera that, like *S. chartarum*, are capable of synthesizing DHN melanin or producing mycotoxins that are dangerous to mammals, such as the genera *Alfaria* and *Myrothecium* [12]. The existence of the sexual state of *S. chartarum* was already postulated by Castellani [47], but for the first time it was fully described based on molecular phylogeny techniques, which revealed that *Ornatispora* and *Melanopsamma* are its teleomorphs [13] and both formed eight-spored asci inside perithecia [12]. These sexual genera are also synonyms of *Stachybotrys* [13]. The vast majority of teleomorphs known to be representatives of this family create perithecial ascomata that are either semi or totally immersed in host tissue. There are characterized by natural resistance to 10% KOH in the presence of which the mycelium structures do not discolor [12]. Both the asexual and the sexual stages of *Myrothecium* are very similar to those known for representatives of the genus *Stachybotrys* (*Melanopsamma*) [47,49]. Furthermore, species of the genus *Myrothecium*, in particular *M. roridum* and *M. verrucaria*, like *Stachybotrys chartarum*, produce secondary metabolites that are highly toxic to cells of other eukaryotes [50,51,52].

## 3. Species Description

*S. chartarum* is usually overgrown by other micromycete fungi, especially of the genera *Aspergillus* and *Penicillium*, so it was often impossible to isolate it from building materials in flooded houses, even if those were very infested with this fungus. To obtain a pure culture of strain, it is best to use media rich in cellulose and low in sugar and nitrogen to compete with other molds typical for indoor air, especially from the genera *Penicillium* and *Aspergillus*. Media containing cellulose as the sole source of carbon are also recommended [10]. Malt extract agar (MEA) and potato dextrose agar (PDA) media (Figure 1A–C) are used commonly in studies on the characterization of *Stachybotrys* spp. [21,35,53]. *S. chartarum* on MEA initially forms gray-white and later gray-black colonies on the surface of which bands of dark gray hyphae form. While the reverse side of the colony remains black all the time, secondary mycelium, most often white in color, may develop on the obverse [54]. Mentioned ability to form dark pigmented structures of mycelium, specifically rich in 1,8-dihydroxynaphthalene (DHN) melanin, is typical for members of the *Stachybotrys* genus with the exception of *Stachybotrys bisby*, which forms hyaline mycelia and conidia and can be found on *Oryza sativa* [55]. Moreover, for *S. chartarum*, a specific feature is the ability to produce extracellular dark pigment on a Czapek yeast extract agar (CYA) and MEA medium, respectively [34,56]. In our case this effect was the best visible on PDA + MEA, with the combination of media used in a ratio of 1:1 (Figure 1C1,C2).

Microscopic features typical for *S. chartarum* anamorph (Figure 2A,B) are rich in melanin with regularly septate hyphae 2 to 4 μm width on which olive-brown or olive-gray conidiophores form, reaching a length of 30–70 μm and a width of 3–5 μm. These are macronematous and mononematous, growing out solely and then erect or formed in groups as sympodially branched conidiophores. A mature conidiophore usually has a rough, slightly papillary surface in its upper part and is darker in color [12,31,57]. Conidiophores are usually septate, with 3–12 phialides radiating from the conidiophore apex. Mature conidiogenic cells are most often club shaped, narrower at the base and extended at the apex, smooth to verrucose and olive-brown, sometimes with slight collarettes. One-celled aseptate conidia are typical ameroconidia and are generated by phialides individually and successively. These phialoconidia initially hyaline and smooth, become dark brown and thick walled with globose to a limoniform or cylindrical shape and sometimes with an apical hilum. Mature conidia are relatively large (7–12 μm × 4–6 μm), usually bearing ornamentations and because of a sticky surface are clustered in heads [12,57]. It is commonly known that conidia are sticky and strongly adhere to surfaces because of a polysaccharide matrix that coats these structures, but its detailed composition remains undiscovered [58]. For this reason, the surface of colonies growing on mycological media as well on walls inside the buildings is wet and tarry black [59] The shape and ornamentation of conidia play a significant role in distinguishing *S. chartarum* strains from *S. yunnaniensis* [33] and *S. chlorochalonata* [34]. Conidia of the former, unlike those typical for *S. chartarum*, are cylindrical to semi-cylindrical. The morphological differences between *S. chlorochalonata* and *S. chartarum* are that the former forms smooth conidia and colonies with limited growth on CYA, and gives off a green extracellular pigment into the medium [15]. In conventional diagnostics the distinction between *S. chartarum* and other species of the *Stachybotrys* genus depends mainly on the shape, color, size and ornamentation of conidia. However, it should be borne in mind that the color and ornamentation of conidia changes with age; therefore, an attempt to identify young colonies lasting a few days may lead to misdiagnosis [13,33,35,37]. *Memnoniella echinata* was considered morphologically similar to *S. chartarum*, even to the extent that the taxonomic distinctions between the genera have been the subject of controversy in the past. On the other hand, these two species are significantly different in terms of metabolomes and spectrum of produced mycotoxins [60,61]. Unlike *S. chartarum*, where the conidia are concentrated in slime heads, in *M. echinata* the conidia are dry and arranged in long chains [38]. Contrary to the earlier statements, it is well known that conidia of *S. chartarum* are poorly adapted for dispersal by airspeeds that prevail inside houses, dwellings or public spaces [62].

The teleomorphic stage of *S. chartarum* typically form eight-spored clavate asci inside subglobose to obpyriform perithecia consisting of a single layer. The mentioned fruit bodies usually occur solely and are rarely in pairs; they are black with a smooth surface. The ostiolar region is papillary but without periphyses. Additionally, thick-walled, septate and erect setae are irregularly distributed over the surface of perithecium. Ascospores or whole asci are released by ostiole, which poses a refractive apical ring. Typically through the ostioles, one-septate cylindrical ascospores are released with verrucose surfaces and with mucoid envelopes on the apices [63,64].

## 4. Biological and Ecological Aspects

*S. chartarum* is a typical saprophytic micromycete that has spread all over the world [11,15]. In natural environments this fungus feeds as saprophyte by decomposing cellulose and other dead plant matter; however, one work reported on its role in soybean invasion [65]. This black mold is able to grow in a range of pH 3.0–9.8; however, the optimal pH for this fungus is in the range of 5.6–6.0. The optimal temperature for growth is in the range of 20–25 °C, although this fungus can grow even at temperature 2.5 °C [17,66]. Moreover, Ochiai et al. [58] showed that none of 21 tested isolates of *S. chartarum* were able to grow at temperature above 37 °C. Although *S. chartarum* is capable of growing over a wide range of pH and temperatures, it is somewhat limited by the relative humidity (RH) of the air and associated with RH a_w_ values. This is one of the key parameters limiting growth indoors. It is well known that growth of *S. chartarum* at room temperature (~25 °C) is possible with an RH value ≥ 93%, and mycotoxin production by this fungus occurs at a_w_ ≥ 0.95 [16,67]. Both these values are higher than those known for other species of molds. Most fungi are able to actively grow at aw ≥ 0.85, while xerophilic fungi, like those of the genus *Aspergillus*, *Penicillium* and *Eurotium*, are able to proliferate at or below a water activity (aw) of 0.85 [68,69]. An additional factor limiting the spread of *S. chartarum* and the colonization of new ecological niches is the limited ability of dissemination of its conidia in the air. This is due to the fact that, unlike *Penicillium* or *Aspergillus*, phialoconidia of this fungus are clustered in slime heads and mature conidia can be dispersed naturally after drying [11]. As it has already been shown experimentally, the release of conidia of *S. chartarum* is positively related to air flow rate, but negatively related to relative humidity [70]. Moreover, insects may play a role in spreading of conidia, as in case of other fungi [71]. However, drying is not a problem, and the fungus can survive unfavorable conditions and its conidia stay viable for years to decades [11]. Even if the water source runs out, *S. chartarum* may continue to propagate because of catabolic reactions, which provide additional water [72].

## 5. *Stachybotrys chartarum* as a Biodeterioration Factor

Indoor mold growth is a consequence of moisture from water damage, water leaks, condensation, water infiltration, improper RH prevailing in premises or flooding [19]. Mold growth begins when water is moistening porous materials for longer than a 48-h period [73]. However, so-called tertiary colonizers like *S. chartarum* require a_w_ > 0.9 and RH > 90% to grow [19] and constant moisture for active growth [20]. *S. chartarum*, due to its outstanding cellulolytic abilities and predisposition to develop on materials rich in cellulose [17,61], such as wood [3,74], fiberboard, gypsum board [75,76], polyurethanes [77], cellulose fabrics [77] and paper [78], is also the object of special attention as a biodeterioration factor [20,75]. *S. chartarum*, like the known species of the genus *Chaetomium*, belongs to the group of fungi often referred to as “soft rot fungi”. These fungi cause changes in wooden building materials resulting in weight and stability losses of wood, albeit to a very limited extent compared to brown rot fungi [75]. *S. chartarum*, highly cellulolytic, is most often isolated from damp, improperly stored straw or hay, on the surface of which it occurs in the form of tarry-black colonies [79,80]. In houses with humidity problems, this fungus is most often isolated from various types of building materials, in particular from damp gypsum boards [76,81,82] and wallpapers [15,34]. Like other molds, it often occurs in the areas of the so-called “thermal bridges” where water vapor condenses [83]. Although it thrives on cellulose-rich materials, it requires constant moisture for 10 to 12 days to start conidiation [66]. The intensive growth of colonies and the production of mycotoxins on moist material occur especially when the material rich in cellulose is also poor in nitrogen compounds [84]. *S. chartarum* as a hydrophilic species, is seen as a tertiary wall colonizer indoors, following the pioneering species, which are xerophilic and able to grow at aw < 0.8 and RH < 80% (especially of the genera *Penicillium* and *Aspergillus*, which are secondary colonizers able to grow at aw and RH in the range of 0.8–0.9 and 80–90%, respectively). The latter group includes such common species as *Aspergillus flavus*, *Aspergillus versicolor*, *Cladosporium cladosporioides*, *Cladosporium sphaerospermum*, *Mucor circinelloides* and *Rhizopus oryzae* [19].

## 6. Harmful Effects on Humans and Animals Related to the Exposure to *Stachybotrys chartarum*

*S. chartarum* was for many years perceived as an insignificant saprophyte [58]. The first cases of disease caused by this species are known from the area of present-day Ukraine. At that time, they were described in 1938 by Russian scientists, and concerned mainly farm animals, among which horses, being non-ruminants and more susceptible to trichothecene-contaminated feed, had the greatest ailments [85,86]. It is well known that LD_50_ in mammals for satratoxins, which are the most toxic among the trichothecenes, is ~1 mg/kg of body weight [87]. The term stachybotryotoxicosis, which primarily refers to mycotoxin poisoning, was used then for this new disease [88,89]. Later, stachybotryotoxicosis was described on numerous farm animals from various parts of the world [13,90,91]. This disease entity was usually related with moldy hay and straw, often leading to mass deaths of animals [80,92]. This toxicosis in case of animals was characterized by symptoms, such as irritation of the mouth, throat and nose, as well as shock, dermal necrosis, hemorrhage, nervous disorder and cardiac arrhythmia, followed, when the disease persists, by lympha-denopathy with fever, leucopenia, agranulocytic anemia and finally death [79,93,94]. In cases of dead animals, autopsies revealed extensive ulcerations along the entire length of the gastrointestinal tract, degenerative changes in various organs and bone marrow [79]. It has also been shown that direct contact with the skin of a living or dead mycelial thallus provoked a dermatosis and also necrosis in a short time [95]. It was also suggested that a similar disease described in Siberia in 1934, characterized by similar changes as described above and significant mortality, was probably also stachybotryotoxicosis [79]. All the cases were related with direct contact of mycelial structures or secondary metabolites of this toxinogenic fungus with the skin and/or mucous membranes. Such contact was usually manifested by a strong inflammatory reaction and necrotic changes. Nevertheless, in all of these cases, active and invasive growth of hyphae was never observed in tissue.

In humans, the health aspects related with fungi of the *Stachybotrys* genus have been poorly described for a long time. Currently, we can find many reports in the literature on the harmful effects related with *S. chartarum*, which also were fatal [9,58,96,97]. So far, all the ailments in humans related with *S. chartarum* were associated with sick building syndrome in wet buildings [9,98]. Idiopathic pulmonary hemorrhage was for the first time reported in Greece [99]. However, *S. chartarum* as the most probable etiological agent of this disease was first isolated in Texas (USA) from the lungs of a child with typical symptoms [100]. Somewhat earlier, from January 1993 to December 1994, as many as 10 cases were described in the area of Cleveland (Ohio, USA) [101]. A total of 37 cases of pulmonary hemorrhage were described in 1993–1998, of which 12 were fatal [9,102]. These unusual cases of pulmonary hemorrhage, occurring mainly in children aged 1–8 months, were mainly manifested by cough with blood [9]. All patients came from damp or flooded homes, from which *S. chartarum* was abundantly isolated [9,61]. Although a study performed at this time by the CDC did not prove that *S. chartarum* was unambiguously responsible for pulmonary hemorrhage [103], studies conducted in other research centers during this period seemed to strongly suggest that this species was the etiological agent of acute pulmonary hemorrhage in the infants [9]. Nevertheless, the fact remains that, in 138 infants living in moldy homes in the United States, pulmonary hemorrhage was identified between 1993 and 1998 [9].

All of these disease symptoms were the result of the fungus’ strong metabolic activity and exposure to its numerous secondary metabolites. The negative impact may not be the same due to the existence of two divergent phenotypes among strains of *S. chartarum* and due to the presence of specific secondary metabolite gene clusters in two mutually exclusive toxin chemotypes [14,35,46].

It is commonly known that toxigenic fungi-produced mycotoxins can accumulate in all the fungal cells [17,66]. These secondary metabolites may accumulate in hyphae, conidiophores, phialides and conidia, which is also the case with *S. chartarum* [104,105]. These compounds may also diffuse into the medium on which the fungus grows [17,66]. Moreover, it was shown that secreted by *S. chartarum* guttation droplets also play a significant role in the exudation of mycotoxins into the air and surface on which this fungus grows [106]. Conidia of this fungus, due to the highest concentrations of mycotoxins they contain, are considered to be the most toxic [66,107]. It should be emphasized that in the case of satratoxins and atranones, these are produced in different amounts depending on chemotype [108] and genotype [46] of *S. chartarum* and conidia are the cells containing the highest concentrations of these mycotoxins [14]. Furthermore, it was confirmed that satratoxins are produced constitutively [107]. It explains why inhalation of mycotoxins and/or conidia released after the mentioned slime heads have dried, especially in the case of people who are constantly exposed to them, may cause serious damage to the lungs [58]. In the air of premises where *S. chartarum* grows on various types of finishing materials, a certain number of viable and dead conidia are always present. The latter, although are not able to grow on culture medium, still have strong allergic and toxic properties [11,17]. It is also commonly known that repeated exposure to dry, unextracted but viable *S. chartarum* conidia can induce pulmonary inflammation, arterial remodeling [109] and immune cell infiltration in mice inhaling them. This phenomenon was not observed for heat-inactivated conidia [110]. As it was observed, at four weeks after exposure, a T-helper cell type 2-mediated response was observed, and after 13 weeks, bronchoalveolar lavage (BAL) fluid was composed primarily of eosinophils, neutrophils and macrophages [109]. It is also worth noting that both dead and alive conidia aggressively irritate the skin and respiratory tract [111]. When conidia of *S. chartarum* enter the digestive system, symptoms such as burning in the mouth, nausea, vomiting, diarrhea and abdominal pain appear [112]. Conidia toxicity seems to be closely related to the concentrations of satratoxins present in them [61]. The ellipsoidal conidia of *S. chartarum*, 7–12 µm by 4–6 µm in size, seemingly appear too large to enter the respiratory tract. However, their aerodynamic diameter is around 5 μm. Achieving such a small diameter is possible because the fibers or ellipsoidal particles orient themselves “longitudinally” in the air, in such a way that their aerodynamic diameter corresponds to this narrower dimension [61]. All the adverse effects of *S. chartarum* on human health are taken together in Table 2.

Concerning pathogenicity of *S. chartarum* and looking through the prism of the fatal or life-threatening cases of pulmonary hemorrhage most possibly caused by *S. chartarum* documented in the literature, it is necessary to answer one of the important questions. What features or virulence factors support the assigning this species in the rank of filamentous pathogenic fungi? At the moment, one of the most urgent issues should be to clarify these virulence factors using in vivo studies performed in animal models. Namely, is *S. chartarum* capable of growing in lung tissue or other solid organs at typical for mammals average body temperature? It is also worthy to consider if all the described fatal cases in the literature [9,21,60,99,100,101,102,103,119,120] were directly related to household exposure to an extremally dangerous mycotoxins, mainly satratoxins, produced by this fungus or related with the active growth of this fungus within infected lung tissue. This should be also specified and supported by photographic documentation if lung tissues observed posthumously were invaded by pigmented, dark, regularly septated hyphae like in typical cases of phaeohyphomycosis. Actually, we know many species of fungi like dermatophytes, which, excluding *T. verrucosum*, are incapable of growing at a temperature ≥ 37 °C, but can invade skin, hairs or nails causing superficial mycoses [121]. In the subject literature, so far we can find only one case report on *S. chartarum* infection of the scalp in an 80-year-old immunocompetent male (with no history of pulmonary mycotoxicosis) [122], which, according to the definition, belongs to the category of superficial fungal infections. It should be noted that this case report is poorly documented from the clinical and laboratory side. First of all, such infection should be confirmed by a positive direct microscopical examination of skin scrapings or a positive histological specimen to prove invasive fungal growth in skin tissue. Moreover, the declared etiological factor has not been molecularly identified and the photographic documentations posted within this publication were borrowed from other sources instead of clinical images of infection in the case of the considered patient. Moreover, as the authors declared, an 80-year-old male was treated with amphotericin B for a year, which, when long-term administered orally or intravenously, usually shows many side effects and is a burden for the patient [123]. A recently published work [124], the first case report on invasive *Stachybotrys chlorohalonata* sinusitis that established a 23-year-old male with refractory acute lymphocytic leukemia, appears to be much better documented. The authors declared invasive growth of the fungus, which was confirmed by histopathology and immunohistochemistry (IHC), and the etiological factor was identified according to molecular biology methods including multi-locus sequence typing. However, it should be noted that out of the three sine biopsy tissues, only the first seems to suggest the presence of *S. chlorohalonata* as the etiological agent of this infection. The second sinus biopsy was negative, indicating that the clinical material tested was microbiologically sterile. Finally, the third biopsy performed showed the presence of *Aspergillus calidoustus* as an etiological factor of invasive fungal sinusitis. This species was previously identified as the closely related *Aspergillus ustus*, but unlike *A. ustus*, *A. calidoustus* grows well at or above 37 °C [125]. Moreover, given the growing number of scientific reports on its role as etiological factor of invasive fungal infections, this species is considered as emerging pathogen [125,126,127]. Taken together, it cannot be ruled out that the causative agent of invasive fungal sinusitis in this case was *A. calidoustus*, and some symptoms were aggravated and more complicated by the presence of *S. chlorohalonata*, for which observed germination of conidia could also be possible and was also described previously in the literature [128]. It cannot be excluded that *A. calidoustus* was the primary and main etiological factor of described invasive fungal sinusitis. This is evidenced by the absence of dark pigmented conidia typical for *S. chlorohalonata* (not present on Figure 5 included in the discussed paper) [124]. Observed hyphal elements with vesicular swellings in the histological slides obtained from the first and third biopsies were brown pigmented, which is also typical for *A. calidoustus* [125]. Finally, genomic DNA extraction was performed each time from the obtained pure cultures and not directly from the clinical material.

On the other hand, in accordance with the postulates of Koch, each etiological factor of infection should be isolated and able to develop fully symptomatic disease in another healthy susceptible host [129]. To the best of actual knowledge on the subject of *S. chartarum*-related diseases, the aforementioned Koch postulates seem impossible to meet for many reasons. Knowing that this fungus is easily cultivable in vitro, it would seem that every disease entity most likely related to *S. chartarum*, and especially those like pulmonary hemorrhage, can be clearly confirmed by isolating the elements of this fungus from tissues. According to the literature [9,39,119], it is commonly known that acute pulmonary hemorrhage and any other complaints related to exposure to *S. chartarum* mycotoxins, VOC and allergenic compounds cannot be interpreted in accordance with Koch’s postulates. It is well known that some of the disease entities undeniably related to *S. chartarum* can be induced by dead conidia and other mycelial structures [11,17], which cannot give rise to new colonies. Moreover, many symptoms accompanying the disease entities caused by *S. chartarum* are simply related to mycotoxins [3,8,102,117,130] and also can be associated with hyper inflammation [131]. Thus, isolation of *S. chartarum* will not be possible and Koch’s postulates cannot be met.

The recent study from 2007 [62] shows that *S. chartarum* is poorly adapted for dispersal by airspeeds that prevail in the indoor environment opposite to *Aspergillus fumigatus*, the most common etiological factor of aspergillosis [132]. While the last mentioned conidia can be easily dispersed because they are tiny (ca 2.5–3.5 µm in diameter) and capable of entering the alveoli [132], only modest numbers of conidia will become airborne in the case of *S. chartarum*, even in heavily contaminated homes [62]. Moreover, in the context of *S. chartarum* having the potential to invade the lung tissue, it was recently shown that this species is not able to grow well enough at 37 °C, and this is the maximum temperature for growth of this fungus [58]. It was demonstrated in this study that none out of the 21 isolates tested were able to grow at 38 °C or higher [58]. The authors observed that none of the conidia in the lungs were able to germinate, which allowed them to conclude that *S. chartarum* has no possibility to invade lung tissues. So far, only one paper has described growth of *S. chartarum* in the lungs of rats. Nevertheless, this could be possible because of the immunological status of 4-day-old rats, which in this case would have been immature, hence the inflammatory effect was not developed and germination of conidia in lung tissue was possible [128]. Anyway, as the authors of this work finally stated, *S. chartarum*, even in the immature rat puppies, was not able to establish an effective infection [128]. So, what constitutes the serious risk for humans and animals with long-term exposure to the black fungus *S. chartarum*, even though it is not similar to typical pathogenic black filamentous fungi? Depending on the strains, *S. chartarum* is capable of producing various types of secondary metabolites, in particular trichothecenes, including the following: trichoverroidin derivatives, roridin E and L-2; satratoxin F, G and H; isosatratoxin F, G and H; verrukarin B and J; verrukarol; trichoverroid; trichoverol A and B; or trichoverrin A and B, etc. [17,89]. It is well known that, so far, all the tested *S. chartarum* strains are able to produce the immunosuppressive phenylspirodrimanes, and the most cytotoxic macrocyclic trichothecenes are generated only by the genotype S of *S. chartarum* [133]. Moreover, this black mold is characterized by a strong proteolytic activity [118]. It should be emphasized that the most dangerous satratoxins are generally produced in much greater amounts than the other trichothecenes [89,134]. It was shown that satratoxin H is always the main toxin produced during the growth of this fungus on wallpaper and its concentration can reach significantly higher values during the growth on this substrate compared to concentrations achieved during the growth on fir and fiberglass [134]. Moreover, many strains of this species produce A–G atranones, whose influence on human health has not been fully elucidated yet [102,135]. This black mold produces factors inducing acute pulmonary hemorrhage and the very toxic 7-triprenyl phenol-type sesquiterpenoid derivatives, which are tyrosine kinase inhibitors [94]. In turn, the released proteinases, hemolysins and β-glucan, can cause a number of pathophysiological effects. This fungus is also able to produce stachylysin, a hemolytic protein that breaks down erythrocytes [136]. Exposure to *S. chartarum* structures of mycelium is extremely dangerous for newborns less than 6 months of age, typically leading to pulmonary hemorrhage [119,137]. It was also proven that both satratoxins as atranones can induce DNA damage leading to cell death in THP-1 cells [138]. Nowadays, we have more and more evidence to conclude that *S. chartarum* carries the potential to cause a serious acute inflammatory response mainly via PMNs in alveoli and the peribronchiolar space. It is well known that repeated exposure to conidia of this fungus resulted in increased eosinophilic infiltration into perivascular tissues and proximal alveoli [58].

Even with such a large amount of evidence on the extremely harmful properties of this species, *S. chartarum* was not included as a dangerous species on the list prepared in 1996 by the European Confederation of Medical Mycology (ECMM), which classified micromycete fungi according to biosafety levels (BSL) [139]. As noted by other experts in this topic [140], at the same time and on the list drawn up in the same year by the American Industrial Hygiene Association (AIHA) organization, this fungus ranks first among molds producing mycotoxins that are harmful to human health. It was also suggested [140] that due to the danger of mycotoxins in the trichothecene group (satratoxins, atranones and roridin), *S. chartarum* should be placed in the BSL-2 class due to the secondary metabolites secreted by this species. In accordance with the adopted division, this species would be in the same group of fungi as such opportunistic pathogens as: *Aspergillus flavus*, *Candida albicans* or *Acremonium kiliense* [139,140]. On the other hand, it is commonly known that living or working in moldy indoor environments is usually associated with sick building syndrome (SBS) or other ailments. It is possible that many of the so far described cases that pose a threat to life and health related to *S. chartarum* were rather related to building-associated molds, comprising many species that colonize wet or damp building materials [141]. As stated by Miller et al. [96], there is limited evidence that severe lung changes and neurocognitive damage, respectively, can occur from building exposure to *S. chartarum*. Although there are still studies suggesting a significant threat posed by *S. chartarum* to human health and life and even its pathogenicity [11,85,98,128], the clinical significance of this fungus is contested [96,117].

## 7. Antifungal Agents to Eradicate *S. chartarum* from Indoor Environments

To eliminate the adverse effects of mold on human and animal health, contaminated interior areas should be carefully cleaned with anti-molding preparations such as common bleach [142,143]. The process of eradicating of molds includes removing the contaminated building material and then applying a fungicide to the surface that has had any contact with molding areas. Preventive steps have to be taken to eliminate the development of moisture, which is the main factor promoting fungal growth [144]. Fungicides used indoors should be non-toxic, hypoallergenic, odorless and non-volatile. Additionally, in environments that can promote fungal growth, like those where it is difficult to establish or control relative humidity (RH), the chemical agents should provide long-term protection [145]. A frequently recommended mildew remover from indoor spaces is the common household bleach, whose active agent is sodium hypochlorite [142,143]. Sodium hypochlorite is a protein-denaturing agent [146]. It is a common disinfectant that is effective against bacteria and fungi [143,147,148]. Essentially, 2.4% sodium hypochlorite kills mold and they become uncultivable, but importantly, it also reduces levels of allergens produced by spores [142,143]. This ability is also confirmed by its fragmentation of household allergens [149]. Hypochlorite can be used in the homes of asthma sufferers because it does not worsen breathing parameters; on the contrary, by reducing allergens, the symptoms of the disease are also reduced [150]. Commonly available disinfectants are also used to remove mold. Effective preparations are based on quaternary ammonium chloride compounds, non-ionic surfactants, chelating agents or those containing phenol and glutaraldehyde. Hydrogen peroxide (17%) and isopropyl alcohol (70%) are also effective [144,151]. Actually, many mold-remediation guidance documents are available, reviewed in a book published by Committee On Damp Indoor Spaces and Health [152]. Antimicrobial paints are also commercially available. These can even be used on mold-contaminated gypsum wallboard that cannot be replaced despite recommendations. Paints of this type are applied to dry surfaces previously cleaned with water or bleach. Depending on the manufacturer, the main active ingredients in said paints include calcium hydroxide, titanium dioxide, chlorothalonil or diiodomethyl p-tolyl sulfone [153]. Alternative, “natural” methods of mold control are being sought. Plant essential oils are an example [154]. Essential oils are mixtures of volatile compounds synthesized by plants, among which are terpenes and terpenoids as well as aliphatic and aromatic components, and are characterized by low molecular weight [155]. It is also known that extracts of *Thymus vulgaris* have a pronounced antifungal effect on *S. chartarum* [156], but there is also one report on its isolation from the dried tissues of this plant [157], which rather suggests the contamination of harvested herbs and secondary colonization by this toxic mold. Tea tree oil (*Melaleuca alternifolia)* also shows antifungal properties against some black molds including *S. chartarum* [158]. We currently have a lot of information on the antifungal activity of essential oils toward *S. chartarum*, but still the main problem is related with the ability to compare obtained results because of lack of standard protocols for testing in vitro biological activity of plant oils. Additionally, most are laboratory-based and in situ studies, and those performed in indoor spaces are lacking (including environmental factors) [154]. Moreover, it is also worthy to consider whether essential oils will be as cost effective as bleach. Furthermore, another important gap in our knowledge so far is the effects of essential oils on the health of people with sick building syndrome, which could possibly exacerbate the disease symptoms.

## 8. Conclusions

The research problem related to this fungus seems to be not overly publicized, and there is still demand to truthfully define the real threats of *S. chartarum* and phylogenetically related species. Although there are currently about 500 publications on *Stachybotrys* spp., there are still some gaps in our knowledge that could constitute important areas for further research. The most important problem, which should be fully elucidated as soon as possible, remains the clarification of the pathogenicity of *Stachybotrys chartarum* and related species, such as *S. chlorohalonata*. Actually, we know that none of the *S. chartarum* strains tested so far were able to grow at a temperature ≥ 37 °C, which makes it even more important to clearly determine whether invasive growth inside the mammal body and tissue penetration by hyphae of this black fungus is possible. Even if it is not possible due to the temperature, which inhibits growth, it should be determined whether superficial infections of the skin, nails, etc. are likely to occur. In the light of the currently available data, it is highly probable that all the changes in internal organs described so far, including gastrointestinal and pulmonary hemorrhage, as well as necrotic changes on the skin or mucous membranes, are simply related to secondary metabolites of this fungus, including mycotoxins. Certainly, the answer to these questions will require appropriate in vivo experiments on suitable animal models, considering both fully immunocompetent individuals as well as in various levels of immunosuppression, or with defects in the immune system. The issue of developing the tests, which will need to be quick, cheap and specific in detection of *Stachybotrys* spp. in dwellings, seems to be equally important. Certainly, many of these issues will be clarified and more light will be shed on these bothering issues as the *S. chartarum* genome is fully annotated and the secondary metabolism pathways and metabolome of this black toxic mold are better known. Last but not least, it seems important to us to draw attention to the need to use more precise terminology in the context of well-documented pulmonary or gastrointestinal hemorrhage cases described in the literature, undeniably related to species of the genus *Stachybotrys*. This seems justified because we currently have a wide range of diagnostic tools to confirm *Stachybotrys*-induced pulmonary hemorrhage (SIPH) or *Stachybotrys*-induced gastrointestinal hemorrhage (SIGH).

## Figures and Tables

**Figure 1 biology-11-00352-f001:**
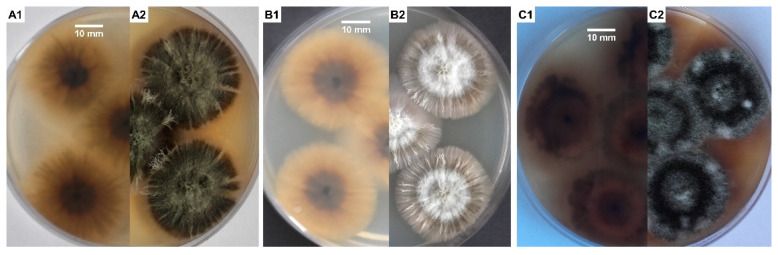
According to literature [12,22,34,53,54,56], these images are typical for *Stachybotrys chartarum* morphology of colony after 10 days incubation at 25 °C on the following: PDA medium (**A1**,**A2**); MEA (**B1**,**B2**); PDA + MEA, with the combination of media in a ratio of 1:1 (**C1**,**C2**), respectively, reverse/averse; strain isolated from gypsum board in flooded home, Cracow, Poland (*S. chartarum* MD1/2021). Photos by the author.

**Figure 2 biology-11-00352-f002:**
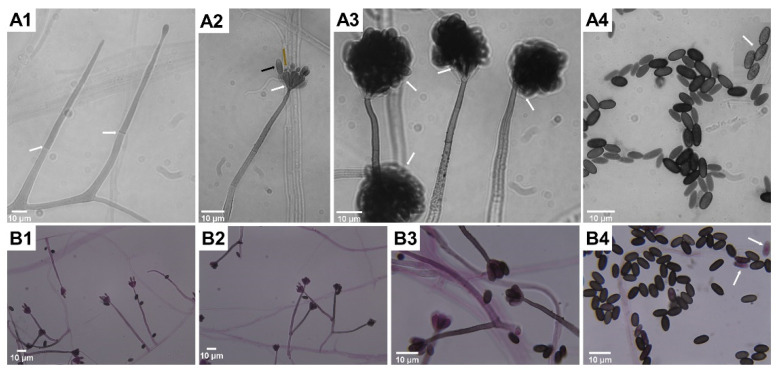
According to the literature [12,13,31,34,35,37,57], these images represent *Stachybotrys chartarum*-specific mycelial structures typical for anamorphs visualized in light microscopy (saline wet mount microscopy, (**A1**–**A4**); fungal structures stained by lactofuchsin, (**B1**–**B4**)). Successive stages of development of mycelial structures: single, erect and septate (white arrows) conidiophores formation (**A1**); forming inverse bottle-shaped phialides (white arrow) with the apex (brown arrow) on top of conidiophore, successively produced single phialoconidia (black arrow, (**A2**)) clustered in slimy heads (white arrows, (**A3**)), which, when dried, release conidia (**A4**) with a rough surface (white arrow). Characteristic for *S. chartarum* formation of sympodially branched conidiophores (**B2**,**B3**), and less often formed solitary conidiophores (**B1**), release mature conidia that are black in color as opposed to immature conidia that easily absorb dye (white arrow, (**B4**)). Magnification 1000× (**A1**–**A4**,**B3**,**B4**) and 400× (**B1**,**B2**), scale bars 10 µm. Photos by the author.

**Table 1 biology-11-00352-t001:** Current taxonomic position of *Stachybotrys chartarum* (Ehrenb.) S. Hughes (1958) and related species ^1−4^.

Taxon Name	Current Name	Synonyms
Kingdom	Fungi	
Subkingdom	Dikarya	
Phylum	Ascomycota	
Subphylum	Pezizomycotina	
Class	Sordariomycetes	
Subclass	Hypocreomycetidae	
Order	Hypocreales	
Family	Stachybotryaceae	
Genus	*Stachybotrys*	*Fuckelina*; *Gliobotrys*; *Hyalobotrys*; *Hyalostachybotrys*; *Memnoniella*; *Spinomyces*; *Synsporium*
Currently accepted species ^1−4^ *S. aksuensis*; *S. albipes*; *S. aloicola*; *S. alternans*; *S. alternans* var. *alterans*; *S. alternans* var. *atoxicus;* *S. asperulus*; *S. atra* f. *genuina*; *S. atra* var. *atra*; *S. atrogriseus*; *S. atrus*; *S. atrus* f. *atrus*; *S. atrus* f. *lobatus*; *S. atrus* var. *atrus*; *S. atrus* var. *brevicaulis*; *S. atrus* var. *cylindrosporus*; *S. atrus* var. *microsporus*; *S. aurantius*; *S. bambusicola*; *S. biformis*; *S. bisbyi*; *S. breviusculus*; *S. cannae*; *S. chartarum*; *S. chlorohalonata*; *S. clitoriae*; *S. cordylines*; *S. crassus*; *S. cylindrosporus*; *S. dakotensis*; *S. dichrous*; *S. dolichophialis*; *S. echinatus*; *S. elasticae*; *S. elastus*; *S. elegans*; *S. elongatus*; *S. eucylindrospora*; *S. freycinetiae*; *S. frondicola*; *S. gamsii*; *S. globosus*; *S. gracilis*; *S. guttulisporus*; *S. havanensis*; *S. humilis*; *S. indicoides*; *S. indicus*; *S. jiangziensis*; *S. kampalensis*; *S. kapiti*; *S. klebahnii*; *S. leprosus*; *S. levisporus*; *S. limonisporus*; *S. littoralis*; *S. lobulatus*; *S. lobulatus* var. *lobulatus*; *S. longisporus*; *S. longistipitatus*; *S. lunzinensis*; *S. magniferae*; *S. mexicanus*; *S. microsporus*; *S. mohanramii*; *S. musae*; *S. nepalensis*; *S. nephrodes*; *S. nephrospora*; *S. nielamuensis*; *S. nilagirica*; *S. oenanthes*; *S. oleronensis*; *S. pallescens*; *S. palmae*; *S. palmicola*; *S. palmijunci*; *S. papyrogena*; *S. parva*; *S. parvispora*; *S. *phaeophialis**; *S. proliferata*; *S. pulchra*; *S. punctatus*; *S. queenslandica*; *S. ramosa*; *S. reniformis*; *S. renispora*; *S. renisporoides*; *S. reniverrucosa*; *S. ruwenzoriensis*; *S. saccharii*; *S. sansevieriae*; *S. sansevieriicola*; *S. scabra*; *S. setosa*; *S. sinuatophora*; *S. socia*; *S. sphaerospora*; *S. stilboidea*; *S. subcylindrosporus*; *S. subreniformis*; *S. subsimplex*; *S. subsylvaticus*; *S. suthepensis*; *S. taiwanensis*; *S. terrestris*; *S. thaxteri*; *S. theobromae*; *S. thermotolerans*; *S. variabilis*; *S. verrucispora*; *S. verrucosa*; *S. virgata*; *S. voglinii*; *S. waitakere*; *S. xigazenensis*; *S. yunnanensis*; *S. yushuensis*; *S. zeae*; *S. zhangmuensis*; *S. zingiberis*; *S. zuckii*		*Stachybotrys chartarum* so far known by 132 synonyms (source: MycoBank [29])

^1^ based on Species Fungorum website [30]. ^2^ according to Lombard et al., Persoonia 2016, 36, 156–246 [12]. ^3^ based on MycoBank website [29]. ^4^ according to Wang et al., Fungal Diversity (2015) 71, 17–83 [13].

**Table 2 biology-11-00352-t002:** Negative aspects related to the presence of *S. chartarum* in homes and dwellings *.

Mycotoxins	MVOCs #	Allergens	Ailments Related with Direct or Indirect Exposure to *S. chartarum* ^
❖ atranones and dolabellanes (chemotype A, about two-thirds of the currently known *S. chartarum* strains); ❖ macrocyclic trichothecenes: - (~39% of isolates), - satratoxins F, G and H - (~35% isolates), roridins E and L-2, - isosatratoxins F, G and H, - verrucarins B and J, - trichoverroids, - trichoverrols A and B - trichoverrins A and B, - trichodermol (chemotype S, about one-third of the currently known *S. chartarum* strains); ❖ phenylspirodrimanes: - stachybotrychromene A-C - stachybotrydial - acetoxystachybotrydial acetate - stachybotrydial acetate - stachybotrylactam - stachybotrylactam acetate - stachybotrysin B and C - stachybonoid D - stachybotryamid - L-671 (*S. chartarum* chemotype A and S);	triprenylated phenolics; trichodiene; acetone; 2-propanol; 1-propanol; 2-metyl-1-propanol; 1-butanol; 2-butanol; 2-methyl-3-buten-2-ol; 3-methyl-1-butanol; 3-methyl-2-butanol; thujopsene; 2-ethylhexanol; 2-ethylhexyl acetate; methyl benzoate; C15 RI1485 13-farnesene; C15 RI 1513 α-curcumene; C15 RI 1519 β-bisabolene; C15 RI 1544 trichodiene; C15 RI 1545 cuparene; sesquiterpenes; 2-ethylhexanol; 3-methylfuran; dimethylhexadiene; dimethyl disulfide; 1-hexanol; 1-octanol; anisole; 2- and 3-methylanisole; sesquiterpene hydrocarbons;	Sta c 3 (21 kDa protein, 144 aminoacids), extracellular alkaline Mg-dependent exodesoxyribonuclease, IgE inducing; 34 kDa unknown secretory protein (SchS34 open reading frame encodes protein of 221 amino acids in length), localized on surface of conidia; stachyrase A (chymotrypsin-like serine proteinase); aspartyl- and metalloproteases; peroxisomal membrane protein; thioredoxin; glutathione reductase; Mn-superoxide dismutase; cyclophilins; heat shock proteins; enolase; alcohol- and aldehyde dehydrogenases; glycosidases; chitin; glycoproteins; β-1,3-D-glucan;	pulmonary hemorrhage **; gastrointestinal hemorrhage **; sick building syndrome (SBS); mycotoxicosis (stachybotrytoxicosis); leucopenia; lymphadenopathy; agranulocytic anemia; asthma; adult nasal and tracheal bleeding; allergies; inflammation; lung injury; pulmonary hypertension; pulmonary arterial remodeling; irritation and necrotic changes within skin and/or mucous membranes; hypersensitivity pneumonitis (repeated inhalation of conidia); neurotoxicity (induction of apoptosis of olfactory sensory neurons (OSNs) in the olfactory epithelium); inhibitory activity against the complement system (K-76-phenylspirodrimane derivative and its oxidation product, K-76 COOH); headache; fatigue; cough; burning nasal passages; tightness of chest; muscle and stomach aches,

* prepared based on the subject literature [13,14,21,25,96,102,113,114,115,116,117,118]; # MVOCs—microbial volatile organic compounds; ^ documented and the most possible; ** considered as related with proteins with emolysin and proteinase activities and stachylysin (hemolysin with hemolytic activity, localized in the inner cell wall of spores and mycelia).

## Data Availability

Not applicable.

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
