# Peer review of "Update on Stachybotrys chartarum—Black Mold Perceived as Toxigenic and Potentially Pathogenic to Humans"

_biology, 2022, doi:10.3390/biology11030352_

Round 1

Reviewer 1 Report

The review was significantly improved. Still, it is strange to have somebody publish a review on a topic they never worked on, but still, the review is of major interest and is well written. The focus is now clear and I only detected minor errors.

Please think about the usage of IPH, because idiopathic are diseases without a known cause. Maybe the authors should rethink the usage in the text. Maybe they should also avoid words like "shocked" in a scientific text.

Table 1 and all sources need careful formatting.

Why do the authors think Stachybotrys only consist of 12 legitimate species? MycoBank counts 126, or am I wrong?

L153-156 needs a reference. Who has shown this in their experiments to be true?

L200: Memnoniella and Stachybotrys are different in their metabolome, just think about the satratoxins.

L230-231: Waht did Ochiai et al. clarifiy?

L302: Do the authors really mean species or maybe genus?

L314-315: This contradicts L81-83.

322-323: When was this published, the publication referenced in the next sentence describes something totally different.

L327-328: Probably the authors don't mean toxic spores, but mycotoxins in spores that are toxic?

L332: inhalation, in L240 the authors describe spores of this fungus not to become easily airborne, so the authors contradict themself in this case. Humans inhale the mycotoxins and probably not the spores.

L438: The authors should add an "e.g." since satratoxins and verrucarins are trichothecenes. 

L446: Grammer

Author Response

Reviewer 1

Comments and Suggestions for Authors

The review was significantly improved. Still, it is strange to have somebody publish a review on a topic they never worked on, but still, the review is of major interest and is well written. The focus is now clear and I only detected minor errors.

Please think about the usage of IPH, because idiopathic are diseases without a known cause. Maybe the authors should rethink the usage in the text. Maybe they should also avoid words like "shocked" in a scientific text.

Our Response

Done, the term "shocked" was replaced by "The first mentioned was of major concern to the public”

We totally agree with Reviewer 1 that term ,,Idiopathic" suggests disease without a known cause. For this reason throughout the manuscript, the term "idiopathic" was removed. Moreover, due to the fact that nowadays the world of science has found an explanation for these cases of pulmonary hemorrange described in the literature, we considered it appropriate to use the term "Stachybotrys induced pulmonary hemorrhage".

Table 1 and all sources need careful formatting.

Our Response

To the best of our knowledge, it was done

Why do the authors think Stachybotrys only consist of 12 legitimate species? MycoBank counts 126, or am I wrong?

We agree with the Reviewer that according to MycoBank website currently we have more legitimate species, exactly 123 different species within Stachybotrys genus. On the other hand only for S. chartarum exist 132 synonyms.

L153-156 needs a reference. Who has shown this in their experiments to be true?

Our Response

As suggested, we cited the correct literature item. Thank you so much for paying attention to this.

L200: Memnoniella and Stachybotrys are different in their metabolome, just think about the satratoxins.

Our Response

Correction was made

L230-231: Waht did Ochiai et al. clarifiy?

Our Response

As suggested by the Reviewer, it was explained

L302: Do the authors really mean species or maybe genus?

Our Response

Of course ,,genus” and correction was made

L314-315: This contradicts L81-83.

Our Response

These sentences have been significantly rewritten and do not lead to misinterpretation at present. We thank the Reviewer for raising this issue.

322-323: When was this published, the publication referenced in the next sentence describes something totally different.

Our Response

We completely agree with the Reviewer that this statement should be made more specific, and that is exactly what has been done.

L327-328: Probably the authors don't mean toxic spores, but mycotoxins in spores that are toxic?

Our Response

An appropriate correction has been made.

L332: inhalation, in L240 the authors describe spores of this fungus not to become easily airborne, so the authors contradict themself in this case. Humans inhale the mycotoxins and probably not the spores.

Our Response

We have clarified this issue and now it should be obvious that young phialoconidia of this fungus are clustered in slime heads while mature conidia can be dispersed naturally after drying.

L438: The authors should add an "e.g." since satratoxins and verrucarins are trichothecenes. 

Our Response

Done

L446: Grammer

To the best of our knowledge correction was made.

Submission Date

19 January 2022

Date of this review

26 Jan 2022 14:27:06

Reviewer 2 Report

Manuscript Title: Update on Stachybotrys chartarum – black mold perceived as toxigenic and potentially pathogenic to humans

Manuscript ID: biology-1578259

The authors reported on Stachybotrys chartarum, one of the most frequently described micromycete fungi in the literature. The review was well organized and the results were properly discussed. However, few major corrections need to be incorporated before being considered for publication.

  1. Abstract should be writtenmore precisely and explain novelty of this work.
  2. Introduction needs precisely with more recent literature.
  3. Please update the Table 1 with more recent reference
  4. Figure 1 and2 need to add the related references.
  5. Provide Table(s)about the mycotoxins, allergens and volatile organic compounds (VOC) as well as the etiological agents of various types of mycoses produced by Stachybotrys chartarum.
  6. There are several mistakesappeared throughout the review manuscript including grammatical errors that need to be fixed.
  7. Please carefully check the referenceformat according to the format requested in the guideline to authors.
  8. Manyof the references have been superceded and more modern ones are required, such as Ann. Mag. Nat. Hist. 1838, 2 (7), 613; Trans. Br. Mycol. Soc. 1943, 26 (3), 133–143; Can. J. Bot. 1958, 36 (6), 727–826. Please carefully check all the references, some parts are missing such as Curr. Opin. Allergy Clin. Immunol. 2005, 5 (2).

Author Response

Reviewer 2

Comments and Suggestions for Authors

Manuscript Title: Update on Stachybotrys chartarum – black mold perceived as toxigenic and potentially pathogenic to humans

Manuscript ID: biology-1578259

The authors reported on Stachybotrys chartarum, one of the most frequently described micromycete fungi in the literature. The review was well organized and the results were properly discussed. However, few major corrections need to be incorporated before being considered for publication.

  1. Abstract should be written more precisely and explain novelty of this work.

Our Response

Abstract was significantly improved

  1. Introduction needs precisely with more recent literature.

Our Response

Introduction has been updated to some extent and supplemented with the latest literature items.

  1. Please update the Table 1 with more recent reference

Our Response

As requested by the Reviewer, Table 1 has been updated based on the latest publications and website, where we can find the most up-to-date information on Stachybotrys species and synonymous names. New sources:

Persoonia 2016; 36:156-246

Wang et al., 2015

MycoBank website

  1. Figure 1 and 2 need to add the related references.

Our Response

Relevant references have been added. Thank you for this idea and in this way we can confirm that the posted photographic documentation is compatible with the macro- and micromorphology descriptions within the manuscript text and commonly available in the subject literature.

  1. Provide Table(s)about the mycotoxins, allergens and volatile organic compounds (VOC) as well as the etiological agents of various types of mycoses produced by Stachybotrys chartarum.

Our Response

Table 2 was prepared in accordance with the recommendation

  1. There are several mistakes appeared throughout the review manuscript including grammatical errors that need to be fixed.

Our Response

Both Reviewers suggest that moderate English changes are required. Therefore, the entire text of the manuscript has been meticulously checked for correctness in English by a native American, and any necessary corrections have been made.

  1. Please carefully check the reference format according to the format requested in the guideline to authors.

  1. Many of the references have been superceded and more modern ones are required, such as Ann. Mag. Nat. Hist. 1838, 2 (7), 613; Trans. Br. Mycol. Soc. 1943, 26 (3), 133–143; Can. J. Bot. 1958, 36 (6), 727–826. Please carefully check all the references, some parts are missing such as Curr. Opin. Allergy Clin. Immunol. 2005, 5 (2).

Our Response

To the best of our knowledge should be: ,,Many of the references have been superseded...” and understanding the Reviewer's recommendation by this way, we introduced appropriate corrections. As suggested by the Reviewer, old references have been supported by more modern ones.

Curr. Opin. Allergy Clin. Immunol. 2005, 5 (2) – we exactly quoted this article within the Introduction section (line 69).

Submission Date

19 January 2022

Date of this review

08 Feb 2022 03:31:37

Round 2

Reviewer 2 Report

The manuscript was improved by the authors.